# Alcohol and Prostate Cancer: Time to Draw Conclusions

**DOI:** 10.3390/biom12030375

**Published:** 2022-02-28

**Authors:** Amanda J. Macke, Armen Petrosyan

**Affiliations:** 1Department of Biochemistry and Molecular Biology, University of Nebraska Medical Center, Omaha, NE 68198, USA; amanda.macke@unmc.edu; 2The Fred and Pamela Buffett Cancer Center, Omaha, NE 68198, USA

**Keywords:** alcohol consumption, ethanol metabolism, prostate cancer, prostate cancer-associated mortality

## Abstract

It has been a long-standing debate in the research and medical societies whether alcohol consumption is linked to the risk of prostate cancer (PCa). Many comprehensive studies from different geographical areas and nationalities have shown that moderate and heavy drinking is positively correlated with the development of PCa. Nevertheless, some observations could not confirm that such a correlation exists; some even suggest that wine consumption could prevent or slow prostate tumor growth. Here, we have rigorously analyzed the evidence both for and against the role of alcohol in PCa development. We found that many of the epidemiological studies did not consider other, potentially critical, factors, including diet (especially, low intake of fish, vegetables and linoleic acid, and excessive use of red meat), smoking, family history of PCa, low physical activity, history of high sexual activities especially with early age of first intercourse, and sexually transmitted infections. In addition, discrepancies between observations come from selectivity criteria for control groups, questionnaires about the type and dosage of alcohol, and misreported alcohol consumption. The lifetime history of alcohol consumption is critical given that a prostate tumor is typically slow-growing; however, many epidemiological observations that show no association monitored only current or relatively recent drinking status. Nevertheless, the overall conclusion is that high alcohol intake, especially binge drinking, is associated with increased risk for PCa, and this effect is not limited to any type of beverage. Alcohol consumption is also directly linked to PCa lethality as it may accelerate the growth of prostate tumors and significantly shorten the time for the progression to metastatic PCa. Thus, we recommend immediately quitting alcohol for patients diagnosed with PCa. We discuss the features of alcohol metabolism in the prostate tissue and the damaging effect of ethanol metabolites on intracellular organization and trafficking. In addition, we review the impact of alcohol consumption on prostate-specific antigen level and the risk for benign prostatic hyperplasia. Lastly, we highlight the known mechanisms of alcohol interference in prostate carcinogenesis and the possible side effects of alcohol during androgen deprivation therapy.

## 1. Introduction

The link between alcohol consumption and malignant diseases has long intrigued researchers. In a review of 140 references from 1966 to 2020, mainly epidemiological studies, moderate alcohol consumption was correlated with increased risk of upper digestive tract, liver, colorectal, breast, pancreatic, and prostate cancers (PCa) [1]. Despite generally increased cancer risk from any type of alcohol, many patients continue to drink alcohol [2], and some governments continue to leave their citizens uninformed of the carcinogenic risk of alcohol consumption [3]. In a recent report, the World Health Organization (WHO) estimated that alcohol use resulted in 3 million deaths globally in 2016, 12.6% of those were associated with malignant neoplasms [4]. Among men, alcohol intake is the fourth-largest contributor to cancer [5].

In 2019, 25.8% of Americans ages 18 or older reported that they engaged in binge drinking in the past month, and another 6.3% reported that they consumed alcohol at a heavy level in the past month [6]. According to the 2019 National Survey on Drug Use and Health (NSDUH), 14.1 million US adults ages 18 and older (5.6% of this age group) had Alcohol Use Disorder (AUD) [7]. This includes 8.9 million men (7.3% of men in this age group) and 5.2 million women (4.0% of women in this age group). Among PCa patients from the 2012–2017 National Health Interview Survey (NHIS), the distribution of alcohol consumption was as follows: 4.2%—heavy drinkers (>14 drinks per week), 47.1%—light/moderate drinkers (up to 14 drinks per week), 12.2%—infrequent drinkers (1–11 drinks in the past year), 25.7%—former drinkers (≥12 drinks in a lifetime but with 0 drinks in the past year), and 10.8%—never drinkers (<12 drinks in a lifetime) [8]. This review summarized the epidemiological, clinical, and biochemical data collected across more than 50 years and aimed to define the role of alcohol in PCa. Here, we discuss publications that either report a link or no link between alcohol consumption and PCa and make some suggestions that may help clinicians stratify patients according to their individual risk profiles.

## 2. The Metabolic Role of Alcohol in PCa

Genetic sensitivity to alcohol’s influence on cancer can be explained by functional variation in alcohol metabolism genes [9]. In a study of cancer-related metabolites, alcohol intake was the 6th leading contributor to metabolite concentration variability in PCa patients, explaining 1.1% of the variability observed [10].

Ethanol (EtOH) is oxidized mainly by cytosolic alcohol dehydrogenase (ADH) to acetaldehyde (ACH). Then, ACH is converted to acetate in mitochondria by aldehyde dehydrogenase (ALDH) (Figure 1). ACH initiates carcinogenesis by forming adducts with proteins and DNA and causing mutations [11,12,13]. In addition, ACH can inhibit O^6^-methylguanine transferase, an enzyme involved in repairing carcinogen-induced DNA alkylation [14]. The action of ACH also is associated with the production of reactive oxygen species (ROS) and reactive nitrogen species, disruption of folate metabolism, and alteration of immune responses [2,15,16]. Thus, it is logical that alterations in this EtOH metabolism may affect an individual’s risk of developing cancer. Esophageal cancer, for instance, has a higher prevalence in populations with a common mutation in ADH [17]. It has been found that the total serum ADH activity was significantly higher in patients with PCa and benign prostatic hyperplasia (BPH) compared to healthy subjects; however, the serum ALDH activity was considerably lower [18]. Interestingly, the same study could not detect substantial differences in ADH and ALDH activity between cancer and BPH groups.

It is a widely accepted concept that, in different cancers, the activity of ADH is disproportionally higher than the activity of ALDH [19]. Thus, the ability of cancer cells to produce ACH surpasses the capability to remove ACH, which would suggest the prolonged exposure of cancer cells to this carcinogen compared to normal cells. However, there are multiple independent reports of high ALDH activity in PCa. Indeed, in prostate tumors, several ALDH isoforms (1A1, 1A3, 3A1, 3A2, 4A1, 7A1, 9A1, and 18A1) have been found to be overexpressed [20,21,22,23,24,25,26,27]. Importantly, the increase in ALDH1A1 expression in PCa was positively correlated with the Gleason score and pathologic grade and inversely associated with the overall survival and cancer-specific survival of the patients [28]. Similar results were obtained from another study that showed that gene expression of ALDH1A1 was increased in patients with high Gleason scores, and ALDH1A1 protein level was higher in advanced-grade PCa than in low-grade and BPH [23]. Next, higher expression of ALDH1A1 was associated with poor survival in hormone-naive patients [29]. The gene expression of ALDH1A3, ALDHB1, and ALDH2 was increased in primary PCa samples compared to BPH samples [29]. In an orthotopic mice model of PCa, PC-3 cells expressing a low level of ALDH generate significantly smaller tumors than their high ALDH counterparts [30]. Similarly, ALDH3A1 was upregulated in lymph node metastases derived from DU145 cells [22]. Different data were found regarding the activity of ALDH1A2. One group found that the expression of ALDH1A2 was reduced in primary prostate tumors compared with the normal prostate [31]. However, recent data from 23,868 men with PCa and 23,051 controls from 25 studies within the international PRACTICAL Consortium indicate that three intronic single nucleotide polymorphisms (SNPs) in the ALDH1A2 gene were associated with longer survival of PCa patients, implying its enhanced activity and quick utilization of ACH [32].

At first glance, increased expression of ALDH enzymes may reduce the contribution of ACH to prostate carcinogenesis. Furthermore, ALDH catalyzes the production of retinoic acid, which was shown to block prostate tumorigenesis [33,34]. However, this still does not rule out ACH accumulation in cancer cells because the rise in ADH activity may be more prominent than that of ALDH. Emerging evidence indicates that different ALDH members can inhibit ROS production induced by chemotherapy or radiation, thus promoting tumor resistance to such therapy [35,36]. Furthermore, in androgen-dependent PCa cells, the ALDH1A3 promoter contains potential AR binding sites, and ALDH1A3 expression is directly regulated by the AR pathway [27].

Excessive alcohol consumption may induce the second pathway of EtOH metabolism: its oxidation in microsomes by cytochrome P450 2E1 (CYP2E1) enzyme (Figure 1). This pathway results in ROS production, which, in turn, may promote oxidative stress and subsequent DNA damage. ROS signaling plays an essential role in the development and progression of prostate malignancy [37]. Results from the Castro group suggested that prostate cytosolic xanthine oxidase can bioactivate ACH to free radicals [38]. A later report from the same laboratory suggested an additional pathway of metabolic activation of EtOH to ACH and 1-hydroxyethyl in the prostate microsomes mediated by flavin-dependent oxidases/peroxidases [39]. They observed ACH accumulation and the signs of oxidative stress in prostate tissue. Moreover, Castro et al. reported that the prostate cytosolic fraction could convert ACH to acetyl radicals [40]. In addition, a recent publication from another group found that the activities of xanthine oxidase and catalase are enhanced in prostate tumor tissue compared to the control healthy tissue (Figure 1) [41]. The authors also found elevated levels of thiobarbituric acid (TBA)-reactive substances and advanced oxidation protein products, the indicators of lipid peroxidation.

Finally, enhanced activity of ALDH enzymes results in excessive acetate production, which can be converted to acetyl CoA. The latter reaction provides an uncontrolled amount of adenosine monophosphate (AMP), the degradation of which is catalyzed by xanthine oxidase and is accompanied by increases in ROS release (Figure 1) [42]. Thus, ACH and free radicals produced in the prostate cells via these alternative pathways may promote tumorigenesis [43,44,45,46].

## 3. Alcohol’s Influence on the Hormonal Status and Prostate Epithelium

One potential link between alcohol and PCa is through the influence of EtOH and its metabolites on the level of testosterone. High levels of circulating testosterone are shown to be associated with an increased risk of PCa [47,48]. The risk was elevated in users of hard liquor and those who had early first intercourse, a higher number of sexual partners, and other indices of high sexual activity [49]. A low alcohol dose results in an acute increase in plasma testosterone [50]. On the contrary, it was suggested that chronic alcohol consumption might increase the metabolic clearance rate of testosterone due to the combined effects of a decreased plasma binding capacity for the androgen and increased hepatic testosterone A-ring reductase activity [51].

Interestingly, a cross-sectional study from four representative provinces of China, including 4535 men, detected higher luteinizing hormone and free testosterone in former drinkers than in never drinkers [52]. A high concentration of total testosterone (TT) was consistently related to alcoholism [53]. Diagonally opposite, acute alcohol ingestion decreases TT levels [54]. Furthermore, chronic alcohol consumption at the moderate and high level has negative effects on endocrine function, resulting in pseudo Cushing’s syndrome [55,56] and the dysregulation of the hypothalamic-pituitary-adrenal axis [57]. Interestingly, men with alcohol-associated liver disease demonstrate hypogonadism, resulting in a high risk of developing BPH [58,59]. Indeed, observations in the rat model of EtOH administration showed significantly decreased serum testosterone level [60,61,62], alterations of the hypothalamus-pituitary axis, and reduction in the weight of the prostate, testicles, and seminal vesicles [63,64].

Overall, these conflicting data imply that alcoholism-associated PCa is not directly influenced by the level of male sex hormones. In support of this, 18 prospective studies that included 3886 men with incident PCa and 6438 control subjects could not validate any associations between the PCa risk and serum concentration of hormones, including testosterone, dihydrotestosterone (DHT), dehydroepiandrosterone sulfate, androstenedione, androstanediol glucuronide, and estradiol [65]. There is no doubt that androgens promote PCa in experimental systems. Still, at this point, there is no convincing evidence that the development of PCa in humans relies heavily on elevated testosterone [66]. However, low levels of sex hormone-binding globulin (SHBG) were correlated with an increased risk of PCa [47]. This is interesting since men who consumed one or more drinks/day had a lower level of SHBG compared with men who drank with less intensity [67].

Chronic and acute alcohol administration in rats results in the following significant alterations in prostate epithelial cells: disorganization of Golgi [68,69], accumulation of lipid droplets, rupture of the microvilli [70], dilatation of the endoplasmic reticulum (ER), significant condensation of chromatin around the nuclear membrane, and increased presence of apoptotic cells [71]. EtOH may accelerate the inflammatory response in rats with nonbacterial prostatitis [72]. Interestingly, the ventral lobe of the prostate from rats exposed to 6.7% EtOH (*v*/*v*) during the prenatal period (day 11–21) demonstrated significant inflammation, epithelial atypia, and an increased number of proliferative cells [73]. When prenatal alcohol exposure was performed alongside treatment with a carcinogen, n-nitrosomethylurea, histophysiological changes appeared in the prostate glands, which may increase the prostate’s susceptibility to developing neoplasia [73].

## 4. Epidemiological Studies Indicating the Link between Alcohol Consumption and the Risk for PCa

There is growing epidemiological evidence from different countries for a connection between alcohol abuse and PCa incidence, as well as high rates of alcoholism diagnosis in PCa cases [74,75,76,77,78,79,80,81,82]. In a worldwide population-based study, alcohol consumption was found by both univariable and multivariable linear regression analysis to be positively associated with age-standardized PCa incidence rates, prevalence rates, and mortality rates [83].

A meta-analysis of studies published before July 1998 found a strong link between heavy alcohol drinking and PCa; however, they could not find reliable statistical data indicating a direct relationship between moderate alcohol consumption and PCa [84]. In the meantime, the study among 7612 Harvard alumni (from 1988 through 1993) found a positive association between moderate intake of liquor (but not wine or beer) and the risk of PCa [85]. In one of the first large US-based studies, in 1996, authors investigated 981 cases of PCa, and found that the risk of PCa increases with the amount of alcohol drunk; the risk was identical for Caucasians and African Americans [79]. Interestingly, the likelihood of PCa is higher among those who first began drinking before age 15 [78]; however, quitting alcohol consumption did not reduce the risk. One study from Australia found that despite around 15% of people quitting after diagnosis with diabetes, Parkinson’s disease, stroke, depression, or breast cancer, there was no significant association with quitting in patients diagnosed with PCa or melanoma [86], likely because of the uncertainty of risk for alcohol-associated disease progression.

Meta-analysis of 340 studies from the PubMed base (published up to December 2014) detected a significant dose-response relationship with an increased risk of PCa among low (1.30–<25 g/day), medium (25–<45 g/day), high (45–<65 g/day), and higher volume (>65 g/day) drinkers compared to abstainers [87]. The authors also found a statistically significant dose-response relationship in PCa mortality. This echoes the results from the large PCa patient population in South Korea [88]. A recent review of twenty-nine meta-analyses came to the conclusion that light alcohol drinking was not associated with the development of several malignancies, except for a slight increase in melanoma, breast cancer, and PCa [89]. In an article published in 2020, Japanese investigators stated that compared with lifetime abstainers, ever drinkers (at light to moderate levels, those who drank two drinks or fewer per day) demonstrated increased odds for PCa [90]. Importantly, this study categorized patients into six categories by their drink–year levels by multiplication of the daily amount of standardized alcohol use (drinks per day) and the duration of drinking (years): 0 (lifetime abstainer), >0–20, >20–40, >40–60, >60–90, and >90 drink-years. The teetotalers were defined as those who responded that they had never consumed alcohol. A study from the Brazilian National Health Survey (59,791 control and 841 PCa cases) indicates the high incidence of PCa among patients frequently drinking alcoholic beverages five or more days per week [91]. Similarly, a recent (2016) case-controlled investigation from an eastern Algerian population detected a strong positive correlation between PCa incidence and total alcohol consumption [92].

Another prospective cohort study of 294,707 US men aged 50–71 years found that the risk of non-advanced PCa was 25% higher for men consuming ≥6 drinks daily [93]. The follow-up analysis of 101 PCa cases from the Iowa state population showed that, compared with non-drinkers, men consuming alcohol were at increased risk of PCa. The relative risk (RR) parameter was positively correlated with the amount of alcohol consumed weekly, but it is not limited to any specific type of alcohol [94]. Another US-based study found that irregular binge drinking (patients who consumed an average of ≥105 g/week, drinking only on 1–2 days per week) significantly increased the risk of PCa compared with abstainers [95]. Observation of 10,501 PCa cases and 10,831 controls from the NCI Breast and Prostate Cancer Cohort Consortium (BPC3) showed that consuming more than 30 g of alcohol per day (corresponding to two drinks) was associated with an increased PCa risk [96].

Meta-analysis of studies published in the Chinese language showed a significantly higher chance of PCa incidence among drinkers than non-drinkers, with some evidence of a dose-response relationship [97]. Identically, a large Delhi population-based study revealed a higher risk of PCa among current drinkers [98]. Similarly, epidemiological observations from Russia showed that a sedentary lifestyle, obesity, and alcohol consumption are critical factors in the development of prostate malignant tumors [99,100]. A local survey from Catania (southern Italy) also confirmed that higher alcohol intake in addition to a lower use of vegetables, legumes, and fish could be risk factors for PCa [101].

Unfortunately, study observations do not agree on the impacts of various types of alcohol in PCa. One group observed a low risk of PCa with an etiologic fraction attributable to alcohol of 3% [102]. In the prospective cohort of American male health professionals, no clear associations were observed between red wine consumption and the chance of PCa [103]. Indeed, for men who consumed ≤4 glasses of red wine weekly, authors found reduced risk; however, the risk was slightly increased in the group of patients consuming more than four glasses of red wine. It has been assumed that wine’s polyphenols, such as gallic acid, piceatannol, and resveratrol, can antagonize the tumor-promoting effect of ROS due to their antioxidant characteristics [104,105]. Similarly, a study among 58,279 men in Holland and 34,565 men in Washington State (USA) found an increased association of PCa with white wine, but not red wine [106,107]. Meanwhile, another US observation from the same state has shown an increased PCa incidence connected to red wine consumption, with no connection to beer or liquor consumption [108]. Others even found that moderate red wine drinking (but not beer or liquor) was associated with a lower risk of progression to lethal disease [109]. However, the multiethnic cohort study of 84,170 Californian men (1340 PCa cases) failed to find support for the assumption that moderate red wine intake (i.e., 1–2 drinks per day) reduces the risk of PCa [110]. Meta-analysis of 17 observations (published up to December 2017) failed to demonstrate an association between PCa incidence and moderate red wine consumption; however, such a link was detected for patients consuming white wine moderately [111].

A study from Sweden that monitored 256 PCa patients and 252 controls detected a slightly elevated risk for current users of hard liquor [49]. Some studies have found a strong connection between PCa incidence and beer consumption [78,112]. Observation of 156 patients from Uruguay within six years showed an increased risk of PCa associated with beer drinking, implying the possible carcinogenic role of beer N-nitroso compounds in prostate malignancy [113].

## 5. The Impact of Alcohol on the Aggressiveness of Prostate Tumors and PCa-Associated Death

Chronic alcohol consumption has been shown to induce high-grade PCa and metastasis [114,115,116,117,118,119]. One US-based study found that a higher frequency of alcohol intake was associated with increased mortality from PCa [120]. In another survey, Canadian researchers interviewed PCa patients immediately after diagnosis and 2–3 years later to monitor the level of alcohol consumption; mortality data were collected for up to 19 years [119]. They found that patients who consumed more than eight drinks weekly had higher mortality than patients who stopped drinking after being diagnosed with PCa.

Observations from Australian patients indicate that a beer intake frequency of ≥5 days per week was associated with an increased risk of advanced PCa (histological Gleason score eight or higher) [121]. Similarly, an extensive analysis among the Japanese population found a positive association of alcohol consumption with PCa in subjects with advanced disease: compared to non-drinkers, increased risks were observed for those who consumed 0–149 g/week, 150–299 g/week, and ≥300 g/week [122].

A recent study aimed to analyze the link between the total intake of alcohol and death from cancer. Using a large international dataset from the United Nations (UN), Food and Agriculture Organization of the United Nations (FAO), Organization for Economic Co-operation and Development (OECD), World Bank, WHO, U.S. Department of Agriculture, U.S. Department of Health, and Eurobarometer, authors came to the unequivocal conclusion that alcohol is “significantly and positively associated with prevalence and mortality from total, colon, lung, breast, and prostate cancers” [123].

Compared to men without early-life use of alcohol, military veterans with heavier alcohol consumption earlier in life were more likely to be diagnosed with high-grade PCa [124]. Recent US-based analysis also suggested that states with more restrictive alcohol policies had lower alcohol-attributable PCa mortality [125]. Accordingly, the authors stated that “strengthening alcohol control policies may be a promising cancer prevention strategy”. These conclusions are reflected in another US-based study which stated that reducing alcohol intake below one drink per day is associated with a significant decrease in risk of PCa-related death [126]. Similarly, an investigation from Japan found PCa-related death among daily drinkers increased 2.5 times [76].

Significantly, PCa mortality among Mormons and Seventh-day Adventists from Germany, who usually consumed alcohol at a low level, is reduced considerably compared with PCa patients from the best German cancer registry (Saarland) [127]. Interestingly, countries with the lowest mortality rate for PCa (Bhutan, Nepal, Bangladesh, North Korea, Turkmenistan, Uzbekistan, Sri Lanka, Tajikistan, and Yemen [128]) are also characterized by the lowest levels of alcohol consumption [129]. Another recent study analyzed PCa mortality data obtained from 71 countries through the International Agency for Research on Cancer and the Food and Agricultural Organization of the United Nations. Authors found a correlation between increased cases of PCa-related deaths and total animal fat calories, meat, milk, sugar, alcoholic beverages, and stimulants [130].

A recent Mendelian randomization study from the international PRACTICAL Consortium found that SNP within the ALDH1B1 gene was associated with mortality in men with low-grade PCa [32]. The authors of this manuscript concluded: “Reducing alcohol consumption could slow prostate cancer disease progression.” Another study of 3306 PCa patients with European ancestry from the PCa Consortium aimed to examine alcohol intake and SNPs in the four pathways associated with PCa aggressiveness (angiogenesis, mitochondria, miRNA, and androgen metabolism-related pathways) [131]. Investigators found that excessive alcohol intake significantly impacts PCa aggressiveness within the following genetic subgroups: CAMK2D (calcium/calmodulin-dependent protein kinase II delta, enriched in the cell cycle and the calcium signaling pathway), PRKCA (protein kinase C alpha, the modulator of cell adhesion and transformation), and ROBO1 (roundabout guidance receptor 1, cell migration factor and a tumor suppressor gene). In addition, the authors observed an additional moderate link between alcohol and SNPs of genes associated with PCa aggressiveness. These are HGF (hepatocyte growth factor), PDGFB (platelet-derived growth factor subunit B), SYK (spleen-associated tyrosine kinase), PDGFD (platelet-derived growth factor D), and COL4A3 (collagen type IV alpha 3 chain) [131].

It has been shown that SNPs in the TNF-α and IL-10 genes were associated with an increased PCa risk [132,133]. A recent study from the Indian population (105 PCa cases along with 115 control) found that an increasing percentage of TNF-α and IL-10 haplotypes were found to be positively associated with aggressiveness of PCa and alcohol consumption [134].

## 6. Epidemiological Studies Indicating No Link between Alcohol Consumption and the Risk for PCa

The geographical distribution of epidemiological studies that found a weak or no correlation between alcohol intake and risk of being diagnosed with PCa is relatively narrower than those describing strong association (Figure 2). The latter is validated by more worldwide, South American, Asian, and European studies. Yet, the observations that deny the link deserve close attention.

Using participants in the 1970–1972 Nutrition Canada Survey (NCS), authors detected 145 cases of PCa and found that total alcohol intake was not related to subsequent development of PCa, although very moderate consumption of wine (<10 g per day), relative to no consumption, showed an RR of 1.48 (95% CI 1.05–2.09) [75]. A study from Iowa (USA) based on data from 1050 men aged 65 to 101 years (mean age 73.5, 71 incident cases of PCa) found no association between PCa risk and alcohol consumption, in general, or with specific types of alcohol (beer, wine, liquor) [135]. However, there were no data on diet, and the majority of drinkers were referred to as current drinkers.

Another study from Hawaii monitored 198 incident cases with invasive PCa and concluded that low alcohol drinking (one drink per day) is not associated with the development of PCa [136]; however, it is not clear how the duration of lifetime alcohol consumption was determined. A Canadian investigation from 617 incident cases of PCa and 637 population controls failed to find a strong association of PCa with alcohol consumption, but patients were asked to address their usual intake only to the one-year pre-diagnosis period [137]. Similar results were obtained from another research from Canada that included 1623 PCa cases and 1623 controls aged 50–74, which, however, did not include the history of alcohol consumption [138]. In a case-controlled study in Greece, 320 PCa patients and 246 controls were undertaken. They also could not confirm the link between alcohol drinking and PCa; however, the types of alcohol were not reported [139]. Furthermore, in this observation, the control group included patients from the same hospitals without a history or symptomatology of BPH but treated for other diseases. Whether these conditions were alcohol-related is unknown. Similarly, studies from the USA found no significant elevations or reductions in risk for beer, wine, or liquor; however, the dosage and frequency were not mentioned [140,141]. A large cohort European investigation could not detect any differences in the association with PCa for specific beverage types or a strong association between alcohol consumption and advanced or high-grade PCa [142]. However, the authors detected a higher risk of PCa with four or more drinks per day. They also did not rule out that “some subjects misreported their alcohol consumption, especially those with a high intake”.

The study that included around 13,000 men in Denmark could not detect a strong correlation between amount or type of spirits and risk of PCa development, even among patients consuming alcohol at the highest level (>41 drinks per week) [143]. In this investigation, the authors analyzed alcohol consumption in a rigorous way. The observation time for each subject was the period from a point in time when patients reported the current status of wine, beer, or spirits consumption, until follow-up when PCa was diagnosed. However, there was a significant variation in the follow-up time, which ranged from 4.5 to 22.9 years. The cohort study of 22,895 Norwegian men aged 40 years and above also did not find an elevated risk of PCa among patients drinking alcohol within the previous two weeks (1–4 times—148 cases, and >4 times—40 cases) [144]. The data from South Korean respondents also did not detect an association between alcohol consumption and PCa diagnosis, although the heavy drinkers in this study cohort were primarily young men (mean age, 44.3 years; 20-year follow-up study) with a lower incidence of PCa [145].

Another interesting UK-based observation utilized detailed diary entries of alcohol consumption from 8670 adults, with follow-up data from cancer registries through 2009 [146]. Although there was no significantly raised risk for PCa with any specific categories of alcohol consumption, the authors mentioned that alcohol was only assessed at a single time point (May 1984). Another limitation of this study was a relatively small cohort size, especially when alcohol levels were categorized. Several epidemiological analyses from different nationalities reported a weak or no correlation between drinking and PCa incidence [147,148,149,150,151,152,153]; however, a common drawback of these studies was insufficient information regarding the duration (years) of alcohol consumption. Moreover, some studies monitored alcohol consumption within only a certain period and concluded that recent intake is not associated with PCa risk [108,153,154,155,156].

The investigation of PCa patients from twenty hospitals in the USA included 699 cases and 2041 controls [157]. Researchers came to a similar conclusion, including no association even for the highest reported level of alcohol consumption. It is, however, noteworthy that subjects’ family history of PCa and dietary habits were not available for study. The observation of 238 cases of PCa patients from northern California had similar limitations [158].

Several epidemiological studies were conducted among members of different churches [159,160]. They found no association of PCa incidence with alcohol consumption; however, these patients typically belonged to the low alcohol level consuming groups. A study from the Lutheran Brotherhood Cohort Study of 17,633 white males age 35 and older (149 fatal PCa cases) concluded that increased consumption of beer or hard liquor was weakly associated with an increased risk of PCa; however, the details of dosage were not provided [161].

For many years, the prostate specific antigen (PSA) blood test has been the gold standard in PCa screening: the probability of a PCa diagnosis increases with increasing serum PSA levels. In 1994, the United States Food and Drug Administration (FDA) approved the use of the PSA test in conjunction with a digital rectal exam to test asymptomatic men for PCa. In the USA, despite disputes over its contribution to overdiagnosis, PSA-based screening has affected a dramatic stage shift from mostly incurable to mostly curable disease along with a >53% reduction in the PCa mortality rate [162]. Here, we analyzed the number of epidemiological observations before and after 1995, when the era of active PSA screening began. As shown in Figure 3, the vast majority (71%) of manuscripts published before 1995 described no or weak link of alcohol drinking to the risk of PCa. Conversely, most of the publications after 1995 (77%) detected moderate or strong association of alcohol consumption habit with PCa incidence and mortality. Therefore, we propose that, in earlier observations, the control group could include men with asymptomatic BPH or PCa. The rise in PCa incidences linked to alcohol drinking can be partially ascribed to increased cases with early detection of PCa due to PSA testing.

## 7. Influence of Drinking Alcohol on PSA Levels and Risk of BPH

The aim of a recent UK-based study was to investigate the association of alcohol consumption, from self-reported data, with PSA levels and risk of PSA-detected PCa. By analyzing 2400 PSA-detected PCa cases and 12,700 controls matched on age and general practice, authors found that heavy drinking could cause lower PSA levels and a slight increase in the risk for more aggressive PCa [116]. Interestingly, another study of 212,039 men from the UK Biobank showed that low alcohol intake was associated with a lower likelihood of PSA testing [163]. Observation among 1031 New Zealand men found a negative association between beer consumption and total and free serum PSA [164].

In PCa patients consuming both alcohol and tobacco, serum PSA mean values were higher than in non-consumers; moreover, a statistically significant increase was observed for serum PSA when compared between PCa and control groups considering alcohol and tobacco consumption [165]. Short-term lifestyle changes, including avoiding spicy food, alcohol, coffee, and bike rides, were seen to decrease serum PSA values [166]. However, a study from Croatia showed that an increase in the intensity of alcohol consumption (drinks per week) was correlated with a decrease in serum PSA of healthy individuals [167]. These data are in agreement with the observation from South Korea, which detected that lower alcohol consumption was significantly associated with serum PSA above the cut-off (4.0 ng/mL) [168]. In sum here, we suggest that serum PSA levels should be cautiously interpreted with the patient’s history of alcohol drinking.

The meta-analysis of 19 studies (120,091 men) was aimed to determine the link between BPH and alcohol consumption. Exposure levels were divided into strata of up to 5, 12, 24, 36, and more than 36 g per day. The investigators concluded that, depending on the group, alcohol consumption is associated with a significantly or marginally significantly decreased likelihood of BPH but not of lower urinary tract symptoms (LUTS) [169]. A more recent analysis of 26 articles suggested that moderate/modest alcohol consumption in men is associated with a reduced risk of BPH and BPH surgery and decreased LUTS compared to the non-drinkers. In the meantime, excessive alcohol consumption above the recommended threshold (two drinks or less/day) is associated with worse LUTS [170]. A large Korean population-based study (130,454 men with and without metabolic syndrome) confirmed the preventive effect of alcohol on BPH [171]. A recent observation from China using 8563 participants also concluded that alcohol consumption seemed to play a protective role against BPH occurrence [172]. A survey of 957 Korean men (180 non-drinkers, 389 with drinking-related facial flushing, 388 without facial flushing) in the 40–69 age group revealed that drinkers with facial flushing showed a decrease in the risk of BPH relative to non-drinkers. In the non-flushing group, no significant relationship was observed between the number of drinks and BPH development [173]. However, to the best of our knowledge, there are no studies of alcohol’s effect on BPH in patients with an elevated level of TT. This would be interesting because alcohol administration in mice (7.5 mL/(kg·d)) can significantly promote prostate hyperplasia induced by a subcutaneous injection of testosterone propionate [174].

Another intriguing aspect of alcohol’s link to BPH and PCa is the effect of alcohol on Finasteride, a widely prescribed medication for male pattern hair loss and BPH. Finasteride is an inhibitor of 5 α-reductase, which converts testosterone to DHT [175]. This has been found to have varying effects on PCa risk. In the Prostate Cancer Prevention Trial (PCPT), 18,882 men who received Finasteride or placebo were monitored for PCa prevalence over seven years [176]. While the study found Finasteride prevents or delays PCa occurrence, the risk of high-grade PCa was increased in Finasteride-treated men. After an additional 18 years, there was no difference in overall survival or survival after PCa diagnosis between the placebo and Finasteride-treated groups [177]. The study participants were further evaluated to consider PCa mortality after 20 years [177]. This analysis found no difference in PCa mortality between the placebo and Finasteride-treated groups. These long-term follow-up analyses ease the concern surrounding the initial increased risk of high-grade PCa resulting from Finasteride treatment. What is intriguing is that, according to another PCPT data analysis, in the Finasteride group, heavy drinking was associated with an 89% increased risk of total cancer, a 101% increased risk for low-grade cancer, and a 115% increased likelihood of high-grade cancer [114]. The precise mechanism of alcohol’s interference in Finasteride treatment is still unknown. Another analysis of the PCPT data found similar results concerning Dutasteride, a more potent 5 α-reductase inhibitor [117]: men randomized to this medicine and reporting more than seven drinks per week were 86% more likely to be diagnosed with high-grade PCa.

## 8. The Cellular Mechanisms of Alcohol’s Effect on the Progression of PCa

In recent years, it has become clear that alcohol has multiple damaging effects on intracellular organization, transportation, and structure of organelles [178]. Both acute and chronic alcohol exposure results in disorganization of Golgi and inhibition of both intra-Golgi and Golgi-to-plasma membrane trafficking. This, in turn, leads to intracellular accumulation of newly synthesized proteins and subsequent ER stress [179,180,181,182,183,184,185]. In addition, EtOH and its metabolites alter the activity of different Rab proteins, the conductors of post-Golgi transportation, and endocytosis [186,187,188,189,190,191,192,193,194]. Our laboratory revealed that alcohol-induced Golgi disorganization is associated with impairment of both coat protein I (COPI) and coat protein II (COPII) vesiculation and monomerization of the Golgi scaffold proteins, golgins [180,195]. Dimerization of giantin, the largest golgin, appears to be essential for the post-alcohol recovery of Golgi in hepatocytes [196]. Interestingly, giantin also loses its dimeric conformation in advanced PCa cells, which is associated with severe Golgi fragmentation and translocation of critical Golgi glycosyltransferases from the Golgi to the ER [197,198,199]. Recent investigations indicated that Golgi disorganization is a hallmark of cancer progression [200,201,202,203]. We introduced the concept of an “onco-Golgi”, which postulates that in different types of cancer, including PCa, the activation of many pro-oncogenic and pro-metastatic pathways is caused by the mislocalization of resident Golgi enzymes due to disorganization of this organelle [204].

We recently reported that in androgen-responsive PCa cells exposed to EtOH, Golgi disorganization is triggered by ACH and results in enhanced anchorage-independent growth, adherence, migration, and secretion of PSA [205]. Alteration of the Golgi morphology leads to the translocation of glycogen synthase kinase β (GSK3β) from the Golgi to the cytoplasm, followed by the phosphorylation of histone deacetylase 6 (HDAC6) and activation of the downstream heat shock protein 90 (HSP90)–androgen receptor (AR) pathway [205] (Figure 4A). Therefore, EtOH may accelerate AR transactivation, the driver of prostate carcinogenesis. Significantly, EtOH administration promoted the growth of LNCaP-derived xenograft tumors in mice.

Next, we observed that the effect of alcohol on Golgi in low passage androgen-responsive LNCaP cells mimicked the fragmented Golgi phenotype of androgen-refractory high passage LNCaP and PC-3 cells [206]. Transition to androgen unresponsiveness was accompanied by the downregulation of N-acetylglucosaminyltransferase-III (MGAT3), the enzyme that competes with N-acetylglucosaminyltransferase-V (MGAT5) for anti-metastatic N-glycan branching. Moreover, in low passage LNCaP cells, alcohol-induced Golgi fragmentation resulted in translocation of MGAT3 from the Golgi to the cytoplasm, while intra-Golgi localization of MGAT5 appeared unaffected. We observed that within the same clinical stage, the level of Golgi fragmentation and the shift of MGAT3 from Golgi to the ER were more prominent in alcohol-consuming patients [206].

Alcohol’s tumor-promoting effect is also mediated via ER stress. It is known that neoplastic transformations are frequently accompanied by calcium deprivation, oxidative stress, aerobic glycolysis, and DNA damage [207,208,209]. These factors trigger ER stress and launch the unfolded protein response (UPR) [210], aiming to enhance the proliferation of cancer cells and maintain their metabolic homeostasis. The latter, in turn, should adjust the tumor microenvironment to facilitate its survival and expansion [211]. Moreover, angiogenesis and production of antiapoptotic factors and pro-inflammatory cytokines are largely ascribed to ER stress and UPR [212,213,214]. Several recent studies have implicated ER stress and UPR in the development of PCa and the progression of castration-resistant prostate cancer (CRPC) [215,216].

Considering the facts described above, we recently investigated the link between alcohol-induced Golgi disorganization and one of the UPR branches, activating transcription factor 6 (ATF6), which is known to be activated in PCa cells [215,216,217,218]. Typically, 90 kDa ATF6 translocates to the Golgi to be cleaved by site-1 protease (S1P) and site-2 protease (S2P), releasing the 50 kDa fragment. The cleaved ATF6 moves to the nucleus, where it initiates transcription of the genes involved in the resistance by ER stress and UPR [219]. We utilized in vitro and in vivo models of chronic alcohol consumption. In addition, we analyzed tissue sections from normal prostate and from the PCa patients categories with the same grade and Gleason score: non-drinking patients (who do not drink or drink less than once per month) and patients who regularly consume alcohol at a moderate level (12 oz. beer—5–6 times per week; 3–5 glasses of wine per week; 3.4 oz. of strong liquor—2–3 times per week) or at a heavy level (12 oz. beer—2 or more times per day; 4 oz. glass of wine—1 or more times per day; 3.4 oz. of strong liquor at least once per day) [220]. We found that alcohol-induced Golgi disorganization was associated with monomerization of *trans*-golgin, GCC185, the Golgi retention partner of S1P and S2P. This leads to translocation of S1P and S2P from Golgi to ER, followed by intra-ER cleavage of ATF6, accelerated UPR, and cell proliferation (Figure 4B). The segregation of S1P and S2P from Golgi and activation of ATF6 are positively correlated with AR signaling, different disease stages, and alcohol consumption. Significantly, the depletion of ATF6 retarded the growth of xenograft prostate tumors and blocked the production of pro-metastatic metabolites [220]. Another important observation from our study is that in PCa patients consuming alcohol, cribriform patterns in the tumor area (a clinically significant prognostic indicator for advanced PCa) were larger than in non-alcoholic patients. The intranuclear signal of ATF6 in these patterns was enhanced in patients drinking at a high level compared with that in the control group or among moderately drinking patients.

Recent observation indicates that, in the TRansgenic Adenocarcinoma of the Mouse Prostate (TRAMP) model of PCa, chronic EtOH administration (10% in saline) accelerates the tumor growth and metastasis [221]. In primary prostate tumors, alcohol-treated mice, compared to the control, demonstrate significantly increased expression of nuclear factor kappa B (NF-κB), supporting a proinflammatory effect of EtOH. In addition, matrix metallopeptidase (MMP2), laminin, integrin β1, and fibronectin were also elevated, suggesting the enhanced metastatic potential of EtOH-treated tumor cells. Interestingly, co-administration of EtOH and NF-κB inhibitor parthenolide prevent such effects of alcohol, suggesting the leading role of the NF-κB-mediated pathway in alcohol-associated tumor promotion.

Overall, the results presented above indicate that multiple pathways are involved in the underlying mechanisms of alcohol’s effect on prostate carcinogenesis. However, at this point, there is no clear understanding of how EtOH accelerates the development of CRPC, where the tumor no longer completely responds to androgen deprivation therapy (ADT).

## 9. Alcohol’s Interference with Androgen Deprivation Therapy

PCa patients typically receive hormone therapy. ADT is the first-line treatment for most cases of CRPC in conjunction with surgery or radiation in specific settings. However, alcohol consumption, even at a low dose, may influence therapy outcomes. For instance, acute administration of Leuprolide, a luteinizing hormone-releasing hormone (LHRH) agonist, attenuated the expression of alcohol withdrawal behavior, whereas, on chronic administration, it has attenuated the development of alcohol dependence [222]. Furthermore, EtOH-withdrawal on chronic administration increases marble-burying behavior in mice, which is attenuated by Leuprolide [223]. Alcohol can augment the osteoporosis induced by LHRH agonists, Zoladex and Triptorelin [224,225]. Prolonged ADT, low serum 25-hydroxyvitamin D levels, and a history of alcohol excess are important risk factors for osteoporosis and spinal fractures in men with PCa [226,227]. In addition, the typical side effects of antiandrogens that could be worsened by alcohol can include hot flashes, nausea, stomach pain, diarrhea, constipation, dizziness, weakness, headache, and frequent urination [228].

## 10. Conclusions

Despite the limitations discussed here, the link between alcohol consumption and the development of PCa is strong. Still, PCa development depends critically on other factors, notably diet, smoking, age, race (black men have higher incidence and mortality than white men), physical and sexual activity, stress, obesity, family history of PCa, and chronic prostatitis [229,230,231] (Figure 5A). For instance, the risk of PCa associated with the pro-inflammatory potential of the diet is accelerated in low-to-moderate alcohol drinkers [232]. In addition, alcohol intake was directly associated with PCa risk among individuals with lower dietary fiber intake and low folate intake [233,234].

Next, dietary preferences may cardinally vary according to geographical area. For instance, the Mediterranean diet (high intake of vegetables, legumes, fresh fruit, non-refined cereals, nuts, and olive oil, with moderate consumption of fish and dairy, low intake of red meats, and infrequent use of red wine in low dosage) was associated with a low incidence of PCa and low mortality rate in patients without metastasis [235,236,237].

PCa risk is positively correlated with the number of drinks and frequent episodes of binge drinking. Individuals who have first-degree family members with PCa should consider moderate, infrequent alcohol use. Patients diagnosed with any stage of PCa should consider quitting drinking since, even at a moderate level of consumption, EtOH and its metabolites alone are enough to accelerate tumor growth and enhance the metastatic potential of cancer cells. Under such circumstances, the contribution of other risk factors is negligible (Figure 5B). The same strategy should be employed for PCa patients after prostatectomy.

Several factors may be responsible for the noted discrepancies between studies that showed a positive link between alcohol and PCa risk and those that failed to find such an association. These include varied sample sizes, types of alcohol considered, criteria used for control selection and alcohol history categorization, diet, and inaccurate self-reporting. Studies with a larger sample size allowed for greater statistical power but require a similar increase in complexity to adequately control for the variables discussed here. Moreover, it is critical to evaluate the impact of alcohol in patients having a long history of alcohol consumption, as different studies found that lifetime, but not current, alcohol intake is positively correlated with the probability of PCa development [78,79,234,238]; an increasing number of drinking years increased the risk of PCa [239]. For instance, when a study in Brazil was conducted based on lifetime drinking, a positive link between alcohol and PCa was detected [91]. However, in the observation among Brazilian patients with current drinking status only, the risk of PCa was slightly reduced [102].

Epidemiological studies that aim to investigate the risk of PCa among alcohol-consuming patients cannot consider all contributing factors. The most popular variables matched for the cases and controls in case-control studies were age, race and residency, poverty census enumeration district, family income, tea and coffee consumption, serum vitamin A level, education, physical activity, body mass index, smoking status, marital status, dietary preferences, family history of cancer, use of PSA screening, total lifetime female sexual partners, family income, age of diagnosis, height, total energy, carbohydrates, and linoleic acid [240]. Unfortunately, some studies did not include important exclusions for control groups, such as history of any other neoplasm, prostatectomy, and presence of prostatic diseases confirmed by transrectal ultrasonography or digital examination. This limits the statistical power to detect significant correlations between control and PCa groups. The ideal control group would be those screened for PCa but not histologically confirmed.

Many observations did not consider the sick-quitter effect. There are always patients considering themselves non-drinkers because they have previously quit drinking alcohol due to a non-cancer-related condition. Additionally, as with all studies that rely on self-reporting by participants, misreporting (either intentionally or inadvertently) of the level of alcohol consumption cannot be avoided. There is a chance that high consumers were falsely categorized as moderate consumers, leading to an underestimation of the risks. Misunderstanding of the “standard drink” may also contribute to misreporting. The standard drink (or one alcoholic drink equivalent), according to US National Institute on Alcohol Abuse and Alcoholism, contains roughly 14 g of pure EtOH, which is found in 12 ounces (~355 mL) of regular beer, which is usually about 5% alcohol; 5 ounces (~150 mL) of wine, which is typically about 12% alcohol; 1.5 ounces (~45 mL) of distilled spirits, which is about 40% alcohol. However, retrospective consideration is prone to recall bias. In many cases, especially social drinking, patients can not precisely count the actual amount of alcohol consumed within a week. Therefore, the number of drinks per week may vary within studies, depending on the subjective calculations of patients, which, in turn, may affect the accuracy of statistical analysis in both control and PCa groups.

Extensive international studies are required to evaluate the impact of specific beverages on PCa development, lest we miss the forest for the trees. In particular, there is insufficient evidence to conclude that moderate drinking of wine (either red or white) could slow PCa growth or metastasis. The potential antioxidant effect of wine’s polyphenols can be outweighed by the tumor-promoting mechanisms of EtOH described here. In the meantime, epidemiological data suggest that individuals consuming wine in moderation are at low risk for BPH.

Lastly, serum PSA data should be cautiously evaluated for patients regularly drinking alcohol at a high level because it can be affected by alcohol intake immediately prior to testing and by a general history of alcohol consumption. Additionally, the observed level of male sex hormones is irrelevant to the alcoholism-associated risk and progression of PCa.

## Figures and Tables

**Figure 1 biomolecules-12-00375-f001:**
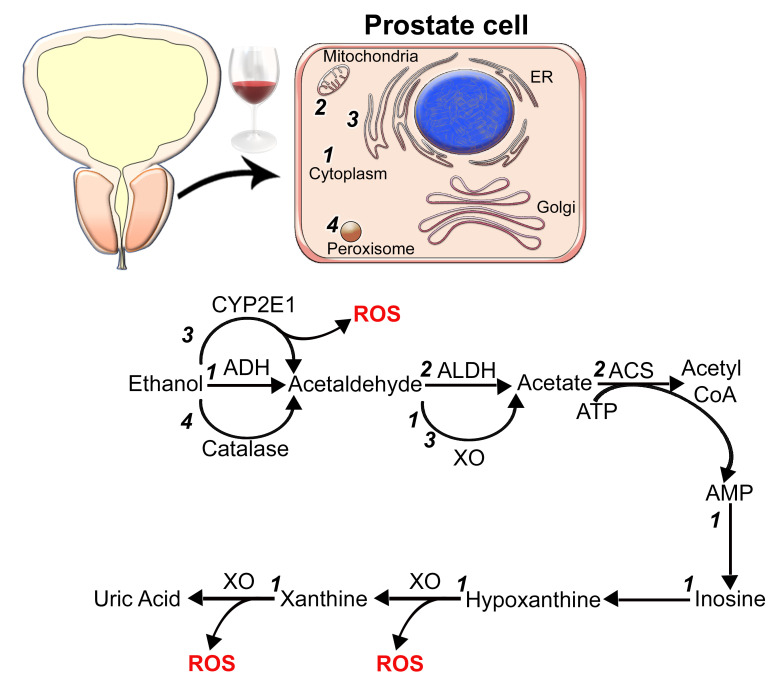
Alcohol metabolism in prostate cells. The main enzymatic breakdown of EtOH is mediated by alcohol dehydrogenase (ADH) and aldehyde dehydrogenase (ALDH). In the cytoplasm (1), ADH metabolizes alcohol to acetaldehyde (ACH), a highly toxic substance and a known carcinogen. In addition, ACH induces endoplasmic reticulum (ER) stress and Golgi disorganization. Then, in the mitochondria (2), acetaldehyde is converted to less active byproduct acetate, breaking down into water and carbon dioxide for elimination. However, chronic alcohol consumption activates alternative pathways of EtOH metabolism. EtOH can be converted to ACH via microsomal cytochrome P450 2E1 (CYP2E1) enzyme (3) or by catalase in peroxisomes (4). The latter reactions are associated with the production of ROS. Excessive production of acetate and accumulation of NADH in mitochondria results in the conversion of ACH into acetate catalyzed by cytoplasmic or microsomal xanthine oxidase (XO). Additionally, acetate can be converted into acetyl CoA (catalyzed by acetyl-CoA synthetase (ACS)) followed by the activation of purine degradation. Oxidations of hypoxanthine to xanthine and xanthine to uric acid are also catalyzed by XO, which contributes to free-radical production. Numbers *1*, *2*, *3*, and *4* indicate localization of enzymatic reactions in the cytoplasm, mitochondria, ER, and peroxisome, accordingly.

**Figure 2 biomolecules-12-00375-f002:**
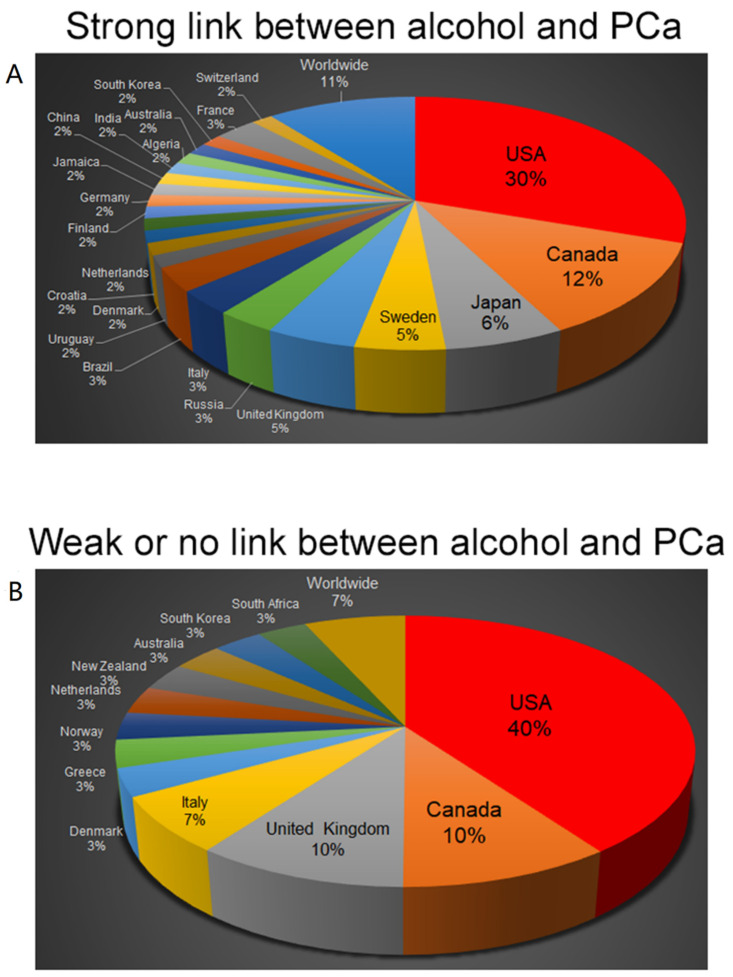
Geographical distribution of the case-control and cohort studies of alcohol association with risk of PCa. Note that epidemiological observations founding strong association between alcohol and PCa risk (**A**) represent a larger number of countries (23) than those denying such a link ((**B**), 13 countries).

**Figure 3 biomolecules-12-00375-f003:**
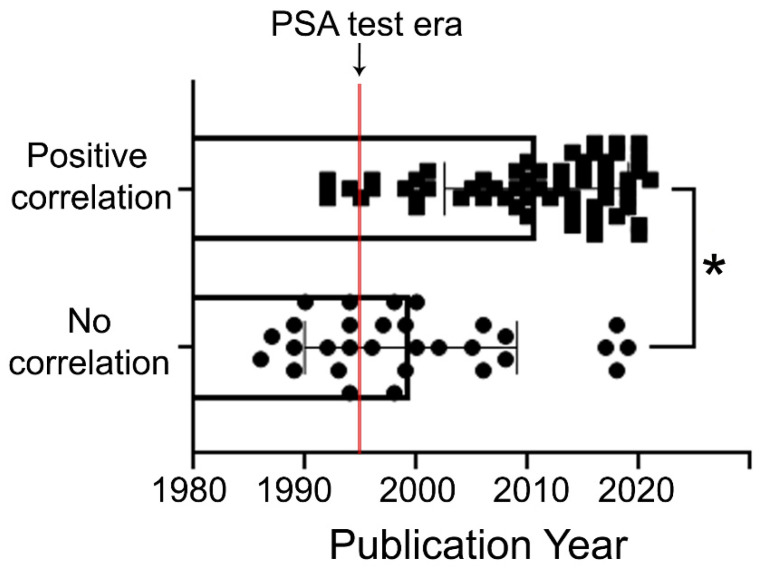
Distribution of epidemiological observations published before and after the implementation of PSA testing in 1995. We found 12 manuscripts published between 1980 and 1995 that deny any link or report a weak association between alcohol consumption habit and risk of PCa. Within these years, only five publications reported the positive connection. In the observations published from 1995 to the present, the ratio of publications showing no link:positive link was shifted to 18:65. There is a significant difference between the publication years of articles with positive correlation and those with no correlation (Mann–Whitney test, * *p* < 0.0001; median ± SD, the median for no correlation is 1998, median for positive correlation is 2014).

**Figure 4 biomolecules-12-00375-f004:**
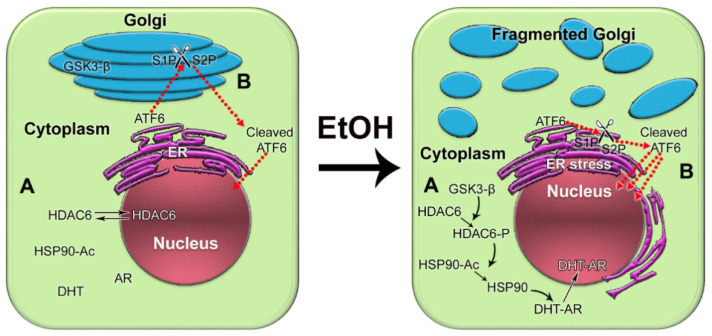
The impact of alcohol-induced Golgi disorganization on prostate carcinogenesis. (**A**) In normal prostate cells or low aggressive PCa cells, HDAC6 is distributed in both the nucleus and the cytoplasm. Typically, the phosphorylation of HDAC6 is moderate because the enzyme that phosphorylates HDAC6, GSK3β, is sequestered primarily within the Golgi. The acetylated HSP90 has a limited binding capacity to AR. EtOH treatment results in Golgi fragmentation and translocation of GSK3β to the cytoplasm, which results in increased phosphorylation of HDAC6. HDAC6-P deacetylates HSP90, which, in turn, accelerates conformational maturation of AR, its binding to DHT, and translocation to the nucleus. (**B**) In normal prostate and low-aggressive PCa cells, ATF6α is cleaved sequentially in the Golgi by S1P and S2P proteases. The dimeric form of *trans*-golgin GCC185 is the retention partner for both S1P and S2P. Cleaved ATF6 enters the nucleus and binds to ER stress-response elements, stimulating the expression of UPR genes. EtOH and its metabolites fragment Golgi membranes, which is associated with the monomerization of GCC185 and the subsequent shift of S1P and S2P to the ER. This simplifies and accelerates ATF6 cleavage, resulting in more prominent UPR signaling to maintain tumor cell growth and proliferation.

**Figure 5 biomolecules-12-00375-f005:**
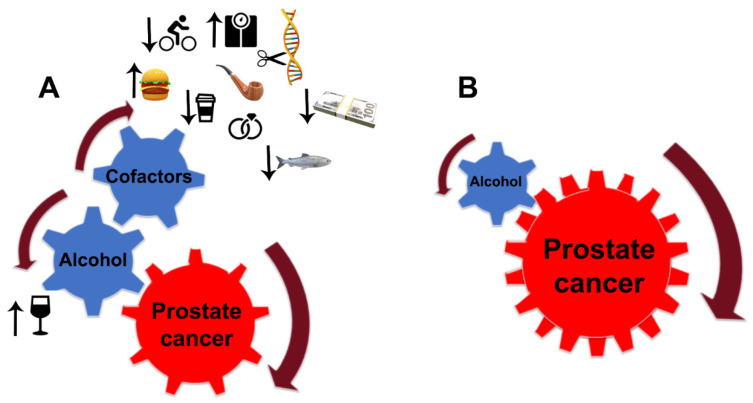
Alcohol interference in the development and progression of PCa. (**A**) Alcohol is a critical player and driver of prostate carcinogenesis. The carcinogenic effects of EtOH and its metabolites are magnified by multiple cofactors, such as obesity, smoking, excessive high-fat and red meat diet, low-level consumption of fish, caffeine, and linoleic acid, low physical activity, SNP of alcohol-related genes, and family history of PCa. Additional factors may include income and marital status: unmarried patients with an unstable financial situation are at higher risk of PCa. (**B**) In patients diagnosed with PCa, alcohol’s contribution to prostate tumor progression does not require cofactors. EtOH metabolites are sufficient to drive tumor growth and raise the metastatic potential of cancer cells.

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
