# Peer review of "Alcohol and Prostate Cancer: Time to Draw Conclusions"

_biomolecules, 2022, doi:10.3390/biom12030375_

Round 1

Reviewer 1 Report

Macke and Petrosyan prepared a comprehensive study on the relationship / effect of alcohol consumption on prostate cancer incidence and severity. The topic is not new and has aroused interest for over 50 years. And yes, my main reservation is that the article is largely a compilation of long-known facts. 60% of the cited literature are items older than 10 years, and only 13% of the cited items were published within the last 3 years. The lack of innovation is therefore a major disadvantage of the current work. Moreover, as relatively new meta-analyses in the subject are available, justification to publish another one rather descriptive in its nature is doubtful.  The most interesting part of the work is the one concerning disorganization of intracellular processes, including the dysfunction of Golgi and enzymes associated with its cisterns – the information not so widely known.

In my opinion, the work should be largely redrafted. Parts based on old reports should be significantly shortened and treated more critically. It would be interesting, for example, to answer the question whether and how the controversy / contradictory data on the alcohol impact on PCa can be linked / evaluated from the point of view of differences in the methodology of the reported studies. Another option is to narrow the scope of the article to the molecular mechanisms engaged in this area and their regulation, or at least to make it the leading part of the article.

Minor remark
Not all the abbreviations are correctly explained (AR, ATF6 among others)

Author Response

We thank the Reviewer for taking the time to evaluate the manuscript. We significantly edited the manuscript according to the recommendations. First, we critically and rigorously analyzed all manuscripts (old and new) that showed no correlation between alcohol consumption and PCa. We explained the significant drawbacks that could impact the evaluation of relative risks. Among them are inappropriate control, low number of samples, lack of information of diet preferences, selection of a specific group of people with an obvious low level of alcohol drinking or PCa incidence. But most importantly, we highlighted in conclusions that most of the observation denying the link between alcohol and PCa lacks information regarding lifetime history of alcohol consumption, focusing instead on current or very recent drinking status. In sum here, we believe that the revised manuscript is not a simple facts statement but a comprehensive novel analysis of all known facts about the link between alcohol and PCa. We have a right to make conclusions as we are the first laboratory that provided the molecular mechanisms of ethanol metabolites' impact on androgen receptor activation and ER stress, the drivers of prostate carcinogenesis. 

Second, we noticed that most of the publications demonstrating the impact of alcohol on PCa incidence or its mortality were published after 1995, when PSA blood testing started to be actively used in most countries. Thus, we assume that the rise of PCa incidences linked to alcohol drinking can be partially ascribed to increased cases with early detection of PCa due to PSA testing. Next, we significantly increased the number of references published within the last ten years; their percentage is more than 50%. The number of publications published in the last three years is 39 now (used to be 26 in the first version)

For the first time, we provide a comprehensive analysis of all known mechanisms of alcohol metabolism in prostate cells and molecular mechanisms that contribute to prostate carcinogenesis. These parts of paper were also significantly edited in text and supplemented by new figures. Also, we provided all abbreviations within and at the end of the manuscript. 

Reviewer 2 Report

The authors prepared a comprehensive review on the effects of alcohol on prostate cancer. An impressive number of papers were included and the manuscript is structurally correct with no major errors.

Only the first sentence of the introduction I find to be a somewhat bipartisan sounding mental shortcut. It is worth pointing out that it is about the effects of alcohol on cancer, the potential mechanism of action, etc.

The excerpts from the work were very enjoyable to read, the chapters are particularly interesting: The metabolic role of alcohol in PCa, Alcohol’s influence on the hormonal status and prostate epithelium,The cellular mechanisms of alcohol’s effect on the progression of PCa

However, part of the chapters like Epidemiological studies, because of the amount of data, it sounded like an enumeration.

A big plus for the neat formulation of the conclusions and the drawings, but here it is worth noting that for such a large amount of data presented, two figures are very few.... especially manuscripts with the form of a review indicate a great potential for graphic design. Here I miss diagrams or tables that would help in reading the text, which is rather monotonous at times. Especially the first chapter could be presented in an interesting way and a summary of epidemiological studies could be prepared in the form of tables. Of course, these are suggestions from the point of view of a reviewer, but it would be nice if the authors had some additional creativity. 

Author Response

We thank the Reviewer for the positive appraisal of our work and for taking the time to evaluate the manuscript. We took into account all recommendations. The first sentence of the Introduction has been modified – it sounds now like: "The link between alcohol consumption and malignant diseases has long intrigued researchers". We added several figures that should facilitate readers' understanding. Also, we tried to avoid simple enumeration of facts and provided critical analysis of the most significant epidemiological observations, especially those that deny the link between alcohol and PCa. We offered new ideas that could explain the contradiction between observations, but we tried to avoid tables because they would be enormous, multiple-page charts that would be hard to follow.  

Round 2

Reviewer 1 Report

I highly appreciate the authors’ efforts to improve their manuscript, which is much better organized and readable now.  Also, supplementing the text with legible figures brings clear advantage. Taking into account the significant social aspect of the analyzed problem, I believe that the article may arouse the interest of a considerably wide audience. Accepting the authors' explanations and their in-depth revision of the manuscript, I believe that it can be accepted for publication in its current form.

Reviewer 2 Report

Authors improved the paper according to suggestions.